# Clock-like Mutation Signature May Be Prognostic for Worse Survival Than Signatures of UV Damage in Cutaneous Melanoma

**DOI:** 10.3390/cancers15153818

**Published:** 2023-07-27

**Authors:** Fabienne Fröhlich, Egle Ramelyte, Patrick Turko, Andreas Dzung, Sandra N. Freiberger, Joanna Mangana, Mitchell P. Levesque, Reinhard Dummer

**Affiliations:** 1Department of Dermatology, University Hospital of Zurich, 8091 Zurich, Switzerland; fabienne.froehlich@usz.ch (F.F.); egle.ramelyte@usz.ch (E.R.); patrick.turko@usz.ch (P.T.); andreas.dzung@usz.ch (A.D.); johanna.mangana@usz.ch (J.M.); mitchell.levesque@usz.ch (M.P.L.); 2Faculty of Medicine, University of Zurich, 8091 Zurich, Switzerland; 3Department of Pathology, University Hospital of Zurich, 8091 Zurich, Switzerland; sandra.freiberger@usz.ch

**Keywords:** clock-like signature, UV signature, COSMIC, mutation profiling, rapid progression

## Abstract

**Simple Summary:**

Although immunotherapy and targeted therapy have dramatically improved melanoma survival, some patients rapidly progress and decease within a few months after a stage IV diagnosis. Up until now, pathological, clinical and biological markers are known prognostic factors for survival of melanoma, while still no prognostic genetic alterations have been identified. Therefore, we aimed to find genetic alterations that predict a short or long survival by sequencing 190 melanoma-related genes of tumor material from 79 patients to contribute to the growing body of knowledge regarding mutational profiling. While no individual gene mutations or combinations of alterations could be linked to overall survival in our study cohort, a clock-like mutational signature according to the Catalog of Somatic Mutations in Cancer (COSMIC) was associated with poor survival whereas a UV mutational signature was prognostic for a longer survival. Those findings are congruent with earlier findings of other authors. Therefore, the prognostic relevance of mutational signatures must be further evaluated in prospective studies.

**Abstract:**

Novel treatment modalities comprising immune checkpoint inhibitors and targeted therapies have revolutionized treatment of metastatic melanoma. Still, some patients suffer from rapid progression and decease within months after a diagnosis of stage IV melanoma. We aimed to assess whether genomic alterations may predict survival after the development of stage IV disease, irrespective of received therapy. We analyzed tumor samples of 79 patients with stage IV melanoma using a custom next-generation gene-sequencing panel, MelArray, designed to detect alterations in 190 melanoma-relevant genes. We classified the patients: first, as short survivors (survival ≤6 months after stage IV disease, n = 22) and long survivors (survival >6 months, n = 57); second, by using a cut-off of one year; and third, by comparing the longest surviving 20 patients to the shortest surviving 20. Among analyzed genes, no individual gene alterations, or combinations of alterations, could be dichotomously associated with survival. However, the cohort’s mutational profiles closely matched three known mutational signatures curated by the Catalog of Somatic Mutations in Cancer (COSMIC): UV signature COSMIC_7 (cosine-similarity 0.932), clock-like signature COSMIC_5 (cosine-similarity 0.829), and COSMIC_30 (cosine-similarity 0.726). Patients with UV signature had longer survival compared to patients with clock-like and COSMIC 30 (*p* < 0.0001). Subgroup dichotomization at 6 months showed that 75% of patients with UV signature survived longer than 6 months, and about 75% of patients with clock-like signature survived less than 6 months after development of stage IV disease. In our cohort, clock-like COSMIC_5 mutational signature predicted poor survival while a UV signature COSMIC_7 predicted longer survival. The prognostic value of mutational signatures should be evaluated in prospective studies.

## 1. Introduction

The treatment of metastatic melanoma has been revolutionized in the past decade. Compared to chemotherapy, which has a median overall survival (mOS) of 9 months [1], novel therapies such as immune checkpoint inhibitors (ICI) or MAPK pathway-targeting inhibitors (MAPKi) have improved 5-year overall survival to 50–60% in metastatic melanoma [1,2,3,4,5,6,7,8,9,10]. However, approximately 20% of patients experience a rapid progression under treatment, independent of therapy type [11,12,13,14,15]. The identification of prognostic factors for survival is therefore of interest to both patients and clinicians.

Currently, known negative prognostic survival markers include clinical factors, such as disease burden, elevated baseline lactate dehydrogenase (LDH) levels, poor Eastern Cooperative Oncology Group (ECOG) performance status, amount of involved organs, presence of brain metastasis, male gender and increased age [16,17,18,19,20], as well as absence of tumor-infiltrating lymphocytes or low tumor mutational burden (TMB) [21,22]. Indeed, immunogenicity of the tumor plays a crucial role in response to ICI treatments: tumors with high lymphocytes and tumor-infiltrating T-cells (TILs) in tumor microenvironment, as well as high levels of anti PDL-1 expression on the tumor, are predictive for response to ICI [3,22]. Further, negative 10-year survival is associated with histopathological findings like increased primary tumor thickness (>1 mm), presence of ulceration and mitotic rate (number of mitosis/mm^2^) [23].

In the recent years, genetic profiling of melanoma patients using Next Generation Sequencing (NGS) has found its way into clinical practice. It is expected to contribute to the adoption of “precision medicine” tailored to the specific needs and genetic profile of individual patients in near future. MelArray is a custom NGS-sequencing panel consisting of >4000 exons across 190 melanoma-relevant genes and is routinely used at University Hospital Zurich. Very recently, genome expression analysis (GEP), through which not only altered genes but also effectively expressed altered genes can be detected, draws attention. Based on this, novel promising therapies like messenger ribonucleic acid (mRNA)-based individualized neoantigen therapy were recently established and probed in clinical trials (e.g., KEYNOTE-942). Those therapies consist of synthetic, patient-specific mRNA coding for neoantigens expressed in the respective tumor [24].

The aim of the current study is to identify possible prognostic genetic markers for the survivorship of melanoma patients using molecular profiles derived from MelArray.

## 2. Materials and Methods

### 2.1. Patients and Samples

In this retrospective, cross-sectional, single center study, we included 79 patients with stage IV melanoma, with available tumor material. All patients were treated at the Department of Dermatology, University Hospital of Zurich, between January 2008 and December 2020, and have consented to using their data for research. Due to the different prognosis than for cutaneous melanoma, we excluded patients with acrolentiginous, mucosal and unknown types of melanomas and metastatic melanoma with unknown primary origin. Due to the small study cohort, no further stratification according to known prognostic survival factors, such as histopathological findings, patient characteristics and/or biological markers (LDH), was performed. Either primary tumor or metastatic tissue was analyzed, depending on the availability. From each patient, one sample was analyzed. Whenever possible, we analyzed metastatic material, since this displays more accurate tumor biology in a metastatic setting. However, in most patients, no metastatic material was available, since invasive procedures are not necessary in the vast majority of cases for the diagnosis of stage IV melanoma.

The main reason for patient exclusion was the lack of available histological material, due to collection before more than ten years (in Switzerland, patient-related data/samples are not stored longer than 10 years). This further explains the clearly higher number of unavailable tumor specimen in the long survivor group.

We classified patients as short survivors (SS), with an overall survival ≤6 months after stage IV diagnosis (22 patients), and long survivors (LS), with an overall survival >6 months (57 patients). The cut-off at 6 months was chosen based on historical OS curves, which flatten at 6 months following a steep drop (e.g., [11,12,13]). In addition, we assessed differences between LS and SS by dichotomizing the groups at one-year survival, and by comparing the 20 longest-surviving to the 20 shortest-surviving patients. Further, to avoid somewhat arbitrary groupings, we assessed survivorship as a continuous function potentially explained by gene alterations and manifestation of different mutational signatures (Figure 1).

### 2.2. NGS Analysis Using MelArray

DNA of tumor tissue was isolated using the Maxwell 16 FFPE Tissue LEV DNA Purification Kit (Promega, Madison, WI, USA) and quantified using a fluorometric assay (Qubit, Thermo Fisher Scientific, Waltham, MA, USA). The KAPA HyperPlus Kit was used to fragment the DNA and build sequencing libraries. Unique sequencing adapters were ligated to the libraries to allow pooling of up to 12 libraries for target capture, which was performed using a customized probe set by Roche NimbleGen (Basel, Switzerland). Batches of samples were sequenced paired-end (150 bp) on one lane of a HiSeq4000 Illumina machine (Illumina, San Diego, CA, USA), resulting in a target sequencing depth of ca. 500 to 1000×.

Raw sequencing data was analyzed using custom bioinformatic pipelines and open-source software. After de-multiplexing, samples were quality controlled using Picard Tools. Reads were then trimmed using skewer v0.22 [25] and aligned using the Burrows-Wheeler aligner (bwa-mem) v0.7.17 [26]. Tools from the Genome Analysis Tool Kit (GATK4) [27] were used to then mark and remove duplicates, perform Base Quality Score Recalibration (BQSR), and detect somatic mutations using MuTect2, according to the Broad Institute’s Best Practices [28]. Putative mutations were annotated using the Variant Effect Predictor version 94 [29]. Variants were then filtered, and a final set of high-confidence non-synonymous variants was prepared per sample.

### 2.3. Statistical Analysis

Gene-level survivorship analysis. We assessed the prognostic power of each individual gene using survivorship analysis. For each gene, patients were divided into “mutated” vs. “wild-type” groups according to whether or not they harbored non-synonymous mutation(s), and the survivorship of these groups was compared using log-rank tests. The p-values from these tests were adjusted using the Benjamini–Hochburg procedure to account for multiple testing. To avoid excessive loss of power from the multiple test correction and to avoid including genes that were rarely mutated and which therefore also lacked sufficient patients to ensure adequate power, tests were limited to the top 30 most frequently mutated genes, which corresponded to being mutated in at least 12 patients. Further, we sought prognostic sets of two genes that may have effects on survivorship when mutated in tandem. For the same genes above, we tested whether double-mutated patients differed in survival time as compared to double-wild-type patients.

Long vs. Short Survivors. We divided the patients into “long” vs. “short” survivors in three ways: first, by using an arbitrary cut-off of 6 months; second, by using a cut-off of one year; and third, by comparing the longest surviving 20 patients to the shortest surviving 20. In each case, the fraction of patients bearing mutations on each gene was compared between LS and SS using Fisher’s Exact Test.

Mutational signatures. We assessed whether we could detect known somatic signatures of mutational processes [30] and whether these signatures may affect survivorship time. To do so, we first extracted bases adjacent to each variant (using Human Genome Reference HG19) to classify each variant as one of 96 possible tri-nucleotide substitutions, and created a matrix of base substitutions. We then performed non-negative matrix factorization (NMF) on the base substitution matrix, using possible numbers of factors (i.e., signatures) ranging from one to ten. The most likely number of signatures present was estimated using Cophenetic correlation, and NMF was again used to extract these signatures from the overall matrix. These extracted signatures were compared to the known COSMIC signatures using cosine similarity. The above steps were performed using the R/Bioconductor package “maftools” [31]. Finally, patients were assigned signatures based on their contribution to each of the extracted signatures, as long as at least 90% of their variants contributed to a single signature. Survivorship analysis was then performed to compare the effects of the different signatures.

## 3. Results

### 3.1. Melanoma-Typical Mutations Are Represented in Patient Cohort

Across all samples, most of the non-synonymous mutations were missense, followed by splice site and nonsense mutations (Figure 2a). The dominating variant type of mutations across cohort was in-frame single nucleotide polymorphisms (SNPs). In-frame oligo nucleotide polymorphisms (ONPs), frame-shifting insertions and frame-shifting deletions were far less common (Figure 2b). The predominant single nucleotide variants (SNV) across cohort were C > T transitions, which are known to be a hallmark for a UV-damage mutational signature (Figure 2c) [32]. Among individual samples, two patients showed a total of >400 mutations, while most patients showed less than 100 mutations (Figure 2d,e). The highest frequency of mutations affected the KMT2D-gene with 125 mutations, followed by BCLAF1. BRAF is at sixth position (Figure 2f).

### 3.2. The 30 Most Frequent Mutations—Overall and Subgroup Analysis

Figure 3 provides an overview of the genomic alterations of the 30 most frequently mutated genes and type of single nucleotide transition (e.g., C > T). Two patients showed impressively high TMB of >400 mutations. One was a short survivor (highest TMB) and the other (second highest TMB) a long survivor. Of the eight patients with the lowest TMB, there were four LS and four SS. The most frequent mutation was BRAF mutation, which was present in n = 46 (58%) of the patients. No significant difference in frequency of BRAF mutations between survival subgroups was observed (Appendix A). The second most frequent mutation was BCLAF1 n = 31 (39%), also without any significant difference between the two subgroups (Appendix A).

### 3.3. Survivorship Analysis

No detected gene alteration was associated with survivorship (log rank tests, adjusted *p* > 0.5, Appendix A).

### 3.4. Long vs. Short Survivors

Comparing the frequencies of mutated vs. wild-type patients for each gene in the LS/SS categories did not reveal any significant associations (Fisher’s exact test, *p* > 0.1). This was true whether the cohort was divided into LS/SS at 6 months or one year, or top vs. bottom 20% survivorship.

### 3.5. Alignment of the Cohorts’ Mutational Profile with COSMIC Signatures

After 6 months, almost 30% of the study patients have died from melanoma (Figure 4a). We first performed single gene analysis of the genes mutated in at least 14 patients and looked for significant differences in overall survival (OS). However, no significant difference in survivorship regarding all evaluated single genes (n = 30) was found (Appendix A). Next, we decomposed the mutational profiles of our cohort and attempted to match them with the known signatures of the Catalog of Somatic Mutations in Cancer (COSMIC). Three mutational signatures from the catalog showed high similarity to our study cohorts’. Concordance was assessed using cosine similarity analysis: COSMIC_7 (cosine-similarity 0.932), known as UV signature, COSMIC_30 (cosine-similarity 0.726) and COSMIC_5 (cosine-similarity 0.829) (Figure 4b). In specific, COSMIC_5 signature was associated with short survival, while COSMIC_7 signature showed an association with long survival (Figure 4c). Patients with COSMIC_7-like signature showed statistically significant, superior OS compared to patients with COSMIC_30-like and COSMIC _5-like signatures (Figure 4d,e, *p* < 0.0001). Furthermore, 75% of patients with COSMIC_7 signature survived longer than 6 months, while about 75% of patients with COSMIC_5 signature survived shorter than or exactly 6 months.

## 4. Discussion

In our study, we performed a custom NGS-sequencing panel, using MelArray, on 79 tumor samples from patients with cutaneous melanoma with the aim to identify prognostic biomarkers for short and long survival after developing stage IV disease. The identified single gene alterations did not correlate with survival. However, we found associations between survival and mutational signatures. Mutational signatures are characteristic combinations of different mutation types, e.g., DNA replication disloyalty, genotoxin exposure and DNA repair and editing defectiveness. The presence of a mutational UV signature was prognostic for longer survival compared to clock-like mutational signatures. UV signatures are defined by high C > T and CC > TT dinucleotide transitions at pyrimidine dimers. Pyrimidines highly absorb UVB radiation. The energy is absorbed by the double bond of the pyrimidine ring and allows the pyridines to react with neighboring molecules. COSMIC_30 and COSMIC_5 mutations are both clock-like mutational signatures, meaning that mutation frequency increases with age in a steady manner. Clock-like, in this context, does not refer to a circadian rhythm and must be distinguished from circadian clock proteins. The opposite of clock-like in our context would be episodic accumulation of mutations over a short period of time.

COSMIC_5 has further been associated with bladder cancer, cancer due to tobacco smoking and NER (nucleotide excision repair) deficiency (COSMIC|SBS5—Mutational Signatures www.sanger.ac.uk (accessed on 26 July 2023)) [34]. There have been recent studies supporting our findings regarding UV signature as prognostic for a beneficial survivorship [35,36].

Assigning of patients to prognostic groups aids in selecting the appropriate treatment and follow-up approach. Clinical stages (e.g., AJCC 8th edition), laboratory parameters (e.g., serum LDH), clinical characteristics (age and male gender) and tumor-specific features are widely used, but cannot independently predict the survival outcome. The cancer immunogram suggests a network of factors to predict the outcomes of patients with cancers [37]. Failure to identify prognostic individual gene alterations (genomic level), suggests that alterations at other levels (e.g., transcriptomic or methylomic) might play a role in disease progression. In 2015, Hugo W. at al reported that up-regulation of gene expression of C-MET, down-regulation of gene expression of LEF1, tumor cell-intrinsic CpG site methylation and YAP1 pathway signature enrichment could be associated with acquired MAPKi resistance [38]. Similarly, differential transcriptomic signatures showed strong association with innate anti-PD-1 resistance in melanoma, but not single gene alterations [39].

Consistent with other studies for prognostic markers in immunotherapy, we found out that patients with tumors that display an enriched UV mutation signature were associated with an improved prognosis, meaning a longer disease-free and overall survival and a better response to immunotherapy [36,40]. Underlying immunological factors in UV signatures showed more putative neoantigens presented via HLA and a favorable immune cell infiltrate of CD4+ memory T cells and M1 macrophages [35]. Further, it has been shown that UV signature is a more reliable prognostic marker than tumor mutational burden [35]. This finding is congruent with our data showing no prognostic benefit towards high mutational burden.

Interestingly, an age-related clock-like mutational signature was associated with poor survival in our study. Recently, Chong W. et al., have associated the same clock-like mutational signature with poor immunotherapy outcome by curating mutational profiles from previous immunotherapy studies of 216 melanoma samples and 113 non-small cell lung cancer samples [36]. Moreover, Chong W. et al., showed that this clock-like signature correlated with lower lymphocyte infiltration and suppressed immune modulation processes. In a novel melanoma cohort, we independently showed that a clock-like mutation signature is prognostic for poor survival, whereas a UV mutational signature predicts improved survival. Our findings emphasize that the assessment of these two signatures may be of use as prognostic biomarkers, if WES panels are applied in patient tumor characterization [36].

Clock-like mutations are understood as a result of environmental/external processes leading to a steady accumulation of mutations and therefore an age-related matter [41]. However, clock-like signatures, like signature 5D, have recently been found by Kim et al., in a study of breast cancer patients to be related to NER and oxidative processes [42], with the latter known as an age-related phenomena. Further, NER pathway among others prevents DNA from oxidative damage. There exists supporting evidence that signature 5 is associated with overload of oxidative stress (either due to exposure or lack of elimination or both) [42].

Limitations of our study comprise a relatively small study cohort of 79 samples. Further, no stratification according to clinical factors besides survival was performed (e.g., age, gender, serum LDH elevation, prior therapies, site of metastasis, amount of involved organs and histopathological features of primary tumor). Strengths of the study are the prior exclusion of patients with uveal and mucosal melanomas and metastatic melanoma of unknown primary origin, which are all suspected to have a different tumor biology from cutaneous melanoma.

## 5. Conclusions

While no gene-specific mutations could be linked to overall therapy-independent survivorship, a clock-like mutational signature was associated with poor survival whereas a UV mutational signature was the best predictive marker for increased survival. Prospective studies are needed to evaluate the prognostic value of our findings and the comparable findings of earlier studies.

## 6. Statement of Translational Relevance

Despite a dramatic improvement of survival with targeted immune therapy, some patients still passed away within months after the stage IV melanoma diagnosis. To date, no biomarkers can reliably predict survivorship. We aimed to find genetic alterations that are associated with survival and potentially unveil new therapeutic targets. We sequenced tumors from short and long survivors and like earlier studies could not identify single gene mutations that correlate with survival. However, we identified mutational signatures, described in the Catalogue of Somatic Mutations in Cancer (COSMIC), which were associated with either poor or beneficial outcome. We found significant associations between clock-like signature, UV signature and patient survival after the development of stage IV cutaneous melanoma. With our findings, we contribute to the collective effort to discover prognostic biomarkers for survival in patients with cutaneous melanoma. If confirmed in prospective studies, these findings may be used for prognostication and may contribute to treatment decisions.

## Figures and Tables

**Figure 1 cancers-15-03818-f001:**
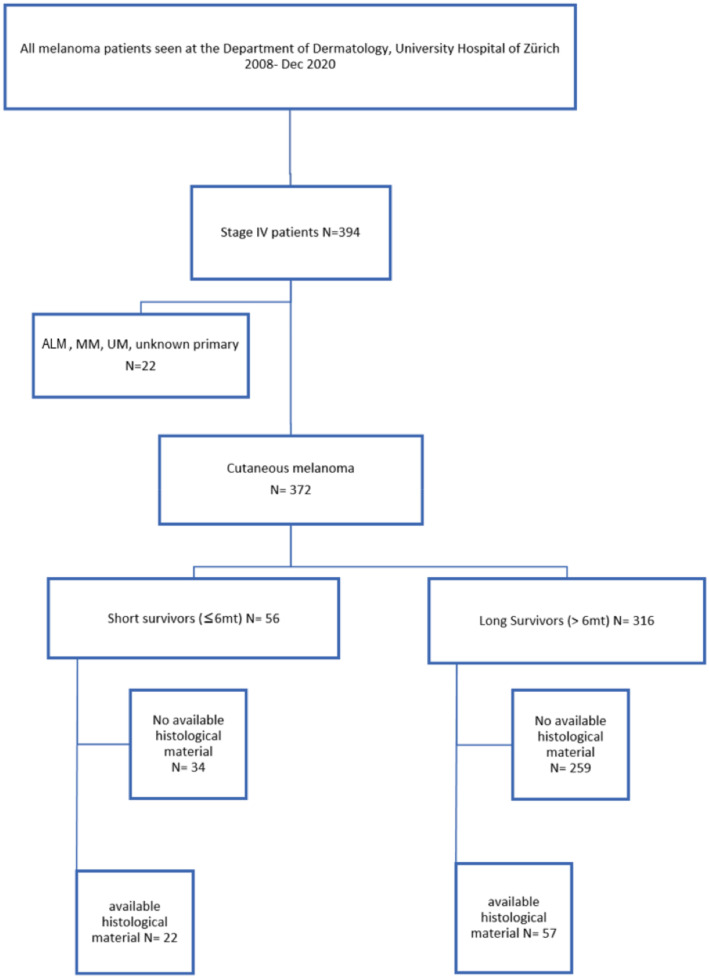
Flow chart showing patients included in the study. ALM—acro-lentiginous melanoma, MM—mucosal melanoma, and UM—uveal melanoma.

**Figure 2 cancers-15-03818-f002:**
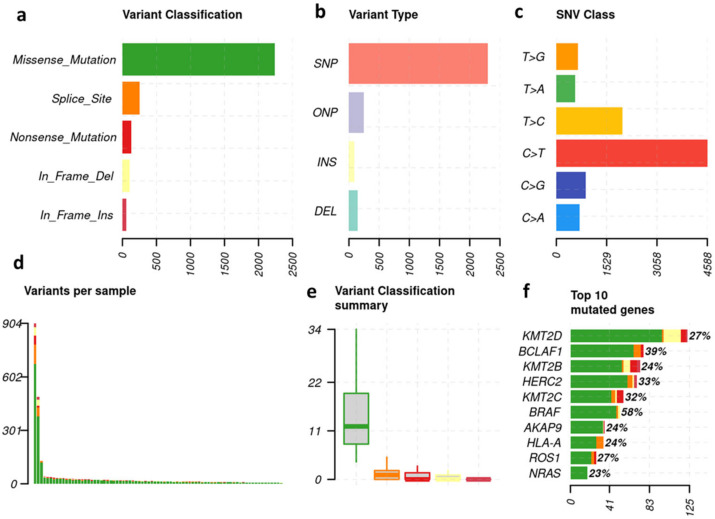
Genetic variant overview in patient cohort. (**a**) Total number of each variant classification detected in the cohort. (**b**) Total number of each variant type detected in cohort. (**c**) Single nucleotide variant (SNV), total number of each nucleotide substitution detected in cohort. (**d**) Total number of variants detected per sample, colored by variant classification. (**e**) Boxplot of the number of variants detected per patient, colored by variant classification. (**f**) Bar length: total number of variants found on each gene, summed across the cohort. Bars are annotated with the percentage of patients who bear a variant on each gene.

**Figure 3 cancers-15-03818-f003:**
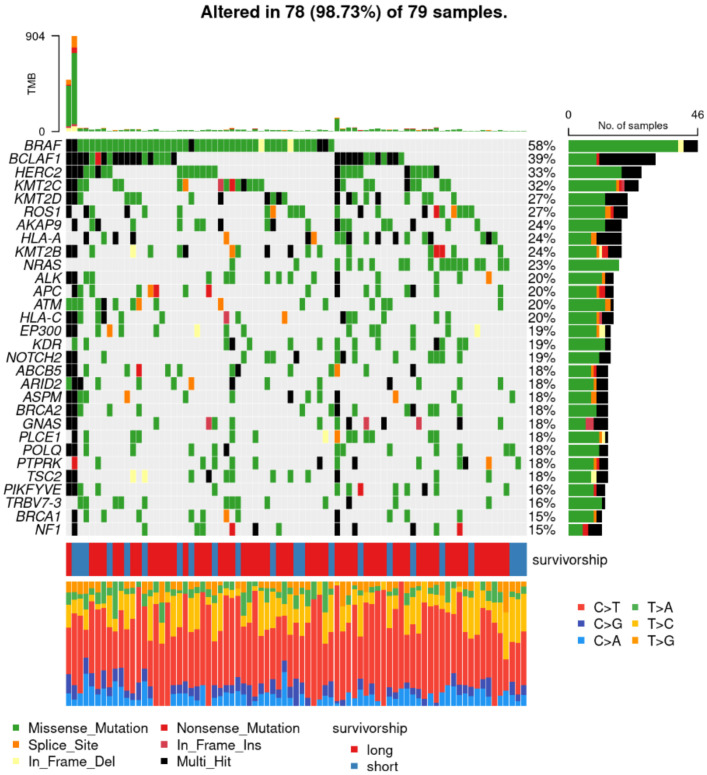
Most common genomic alterations: Rows (y-axis) represent genes and columns (x-axis) represent samples. Different types of genomic alterations, single nucleotide transitions and survival groups are color-coded. The bottom line displays the type of base alteration in the corresponding sample.

**Figure 4 cancers-15-03818-f004:**
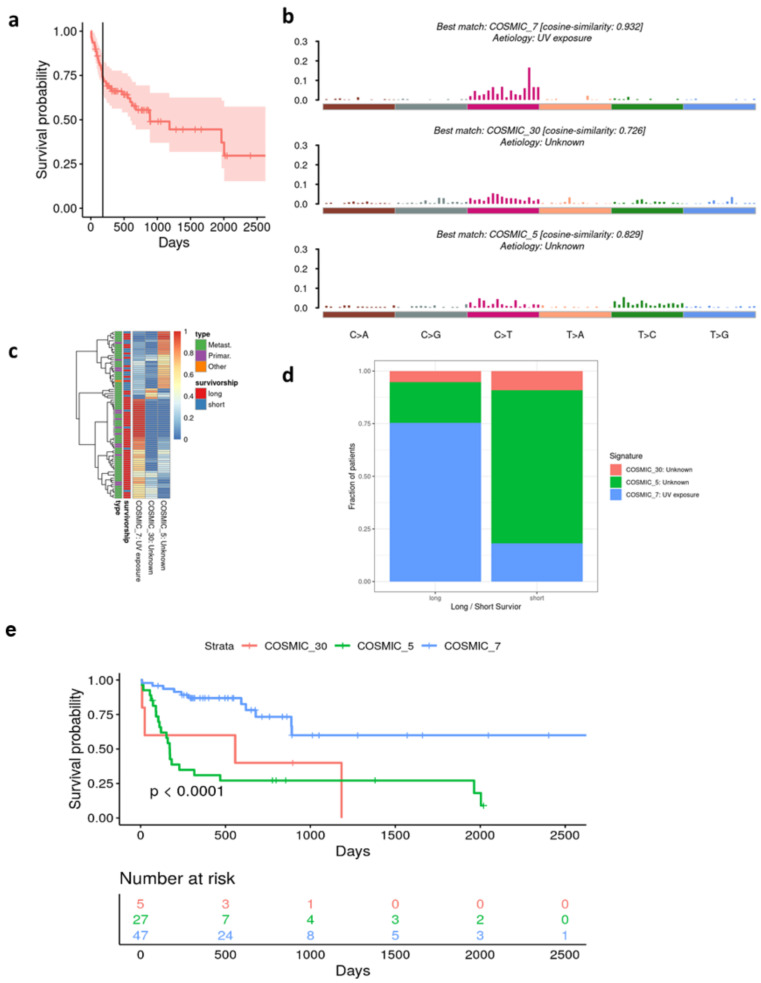
Mutational Signatures. (**a**) Survivorship curve of the whole patient cohort. The cut-off at 6 months (180 days) reflected by vertical line. (**b**) Alignment of mutational signature from our cohort and validated COSMIC signatures [33]. (**c**) Sample-focused heat map indicating sample similarity to COSMIC signatures (1.0 high similarity, 0 low similarity). (**d**) Presence of COSMIC signatures in short and long survivors. (**e**) Survivorship curves of patients harboring different mutational signatures.

## Data Availability

The data generated in this study are available upon request from the corresponding author.

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
