# Peer review of "Clock-like Mutation Signature May Be Prognostic for Worse Survival Than Signatures of UV Damage in Cutaneous Melanoma"

_cancers, 2023, doi:10.3390/cancers15153818_

Round 1
Reviewer 1 Report
Dear Authors,
Interestingly, clock-like mutation signature may be prognostic for worse survival than signatures of UV damage in cutaneous melanoma. Gene signatures and prognostic biomarkers are utterly important, aiding personalized medicine and therapeutic response.
In line 108: Why did you chose 12 patients to test?
In Figure 2C: Please include the meaning of SNV.
In lines 185-186 use 6 months (180 days), as in the figure 4 legend.
In Figure 4a and 4e, correct the x-axis (Days)
In lines 214: replace up-expression and down-expression.
In lines 238-240, 276: use the same font size as the rest of the manuscript.
None
Author Response
Dear reviewer,
many thanks for your valuable inputs. Please find enclosed our detailed answers.
Kind regards,
Fabienne Fröhlich

Reviewer 2 Report
Fröhlich et al performed targeted sequencing of 190 melanoma-related genes on 79 tumors and found prognostic value in assessing mutational signatures, with SBS7 conferring better survival than SBS5.
I have two major concerns:
(i) The sample cohort is small and as such no stratification was done according to critical clinical factors such as age, sex, treatment/therapy, site of the primary, site of metastasis, etc. All these could have a significant impact on the results.
(ii) There is no novelty in the findings. Specifically I don't see what it has to offer over and above what is already known:
· Trucco et al. (Nat Med. 2019 Feb, PMID: 30510256) already showed that UV radiation–induced DNA damage is prognostic for outcome in melanoma using TCGA data of over 400 melanoma patient samples. Furthermore, they identified identify ten recurrently mutated UVR signature genes that predict patient survival and validated their findings in a moue model.
· Chong et al. (Mol Ther Nucleic Acids. 2020 Oct, PMID: 33335795) already showed that the age-related clock-like signature was associated with worse prognosis and lower immune activity in melanoma patients. Furthermore this cohort used WES as well as targeted sequencing and had significantly more patients (n=216 melanoma samples).
· Lotz et al (Br. J. Dermatology. 2021 Feb, PMID: 32282938) model how age-related (clock-like) somatic mutations and extrinsic (ultraviolet radiation) mutations accumulate in melanoma genomes.
· Kim et al., (Front. Genet. 2022 Sept, PMID: 36246650) have already reported that melanomas belonging to the UV-low cluster (SBS1/5 dominant signatures) showed significantly worse overall survival and landmark survival at 1-year than those in the UV-high cluster (SBS7a/7b dominant signature).
A minor point is that the title of the manuscript doesn't exactly sound convincing: “Clock-Like Mutation Signature May Be Prognostic for Worse Survival than Signatures of UV Damage in Cutaneous Melanoma”
Author Response

(The authors gave the same response as above.)

Reviewer 3 Report
This is an interesting and timely study. Although not exactly novel, it adds to the growing body of knowledge around the need to be better able to provide more accurate prognostication for melanoma in the era of Checkpoint blockade. One of the major limitations of the study is its size, and while I am not a statistics experts I would check if the powering of the study is sufficient. I found it clearly communicated and presented. However, I could have liked the authors to probe more critically what is meant by 'clock-like signatures, as for some researchers this can refer to circadian aspects i.e., via the CLOCK proteins. I have some more specific comments which I highlight directly with the authors below.
Author Response

(The authors gave the same response as above.)
